# AUTOCODER: ENHANCING CODE LARGE LANGUAGE MODEL WITH AIEV-INSTRUCT

## ABSTRACT

We introduce AutoCoder, a code Large Language Model that surpasses GPT-4 Turbo `2024-04-09` and GPT-4o `2024-08-06` in pass@1 on the Human Eval benchmark test ($90.9\%$ vs. $90.2\%$). In addition, AutoCoder offers a more versatile code interpreter compared to GPT-4 Turbo and GPT-4o. Its code interpreter can install external packages instead of limiting to built-in packages. AutoCoder's training data is a multi-turn dialogue dataset created by a system combining agent interaction and external code execution verification, a method we term **AIEV-INSTRUCT** (Instruction Tuning with Agent-Interaction and Execution-Verified). Compared to previous large-scale code dataset generation methods, AIEV-INSTRUCT reduces dependence on proprietary large models and provides execution-validated code dataset. The code and the demo video is available in supplementary materials.

## 1 INTRODUCTION

Code generation is a critical aspect of modern software development. It significantly enhances development efficiency and quality by increasing productivity, reducing errors, standardizing code, accelerating prototyping, and supporting complex systems Li et al. (2024; 2023a); Buscemi (2023).Recently, Large Language Models (LLMs), such as GPT-4 OpenAI (2024) and Claude 3.5 Sonnet Anthropic (2024), have achieved significant advancements on code generation. These models have shown high accuracy in producing code that meets user requirements and have been widely adopted in real-world software development.

Training large language models requires extensive high-quality data Hoffmann et al. (2022). This is particularly crucial for code generation tasks that demand high accuracy Chen et al. (2021). OpenAI once hired people to help annotate the Code Instruct dataset for training their InstructGPT Ouyang et al. (2022). However, manually annotating large-scale code instruction datasets is both economically and time-consuming Xu et al. (2022). To address this challenge, previous work has employed various automated code annotation methods, such as SELF-INSTRUCT Wang et al. (2022), EVOL-INSTRUCT Luo et al. (2023), and OSS-INSTRUCT Wei et al. (2023). SELF-INSTRUCT enhances LLMs' instruction-following capabilities by using strong teacher models to generate synthetic coding instructions for fine-tuning weaker student models. EVOL-INSTRUCT improves LLMs' coding abilities by iteratively increasing the complexity of seed code instructions through various heuristics. OSS-INSTRUCT generates diverse and realistic coding problems by drawing inspiration from open-source code snippets. The essence of these methods lies in distilling the knowledge of a powerful teacher model (such as GPT-4 Turbo) to guide a smaller model. This leads to a problem: **While the small model can achieve significant performance improvements, the final accuracy of the small model is unlikely to surpass that of the teacher model.** Because both the correct and incorrect knowledge from the teacher model are transferred to the small model. Moreover, although using closed-source models reduces costs compared to manual annotation, the cost of using closed-source models remains high. According to our tests, even with the relatively cheaper GPT-4 Turbo model, generating an average of 6,500 high-quality entries for the code instruction dataset costs $1,000.

**This raises two questions:**

1. *Can we correct the incorrect knowledge generated by the teacher model to provide more accurate code for the student model?*

2. *Instead of relying on expensive closed-source teacher models, can we enable our student model to learn autonomously?*

To address the **first issue**, we designed a new large-scale code instruction dataset annotation method called AIEV-INSTRUCT. It is an interaction system comprising two agents: a *questioner* and a *programmer*. These agents interact to simulate the process of *programmers* constructing code according to project requirements and conducting unit tests. In each dialogue round, we extract the code generated by the *programmers* and execute it. The execution results are returned to the *questioner* to inform the next round of questions. This process continues until the *programmers*'s code passes the unit tests, ensuring the accuracy of the generated code dataset.

To address the **second issue**, we sperate AIEV-INSTRUCT into two stages: the *Teaching Stage* and the *Self-learning Stage*. In the *Teaching Stage*, we rely on proprietary large models as agents for code annotation, similar to previous methods. Once our model surpasses the proprietary models in accuracy on the test set, we transition to the *Self-learning Stage*. In this stage, we use our own model as the agent for code annotation. For detailed methodology, refer to Section 3.

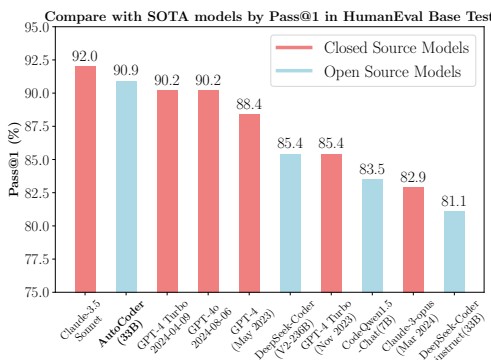

Figure 1: Pass@1 (%) comparison of Various LLMs on the HumanEval Base Test.

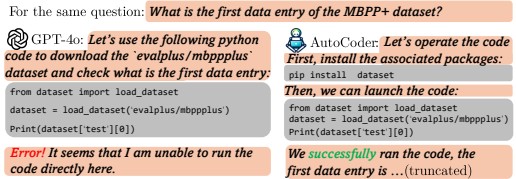

Figure 2: Comparison of Code Interpreter Functions between AutoCoder and GPT-4o. ▨: Nature language generated by the model; ▨: Code generated by the model. AutoCoder can recognize **external package** installation commands, whereas GPT-4o can only run code that includes built-in packages. The demo video is in supplementary materials.

Under the support of AIEV-INSTRUCT, we obtained 169K high-quality code instruction data samples. Using this dataset, we trained the AutoCoder series models, including AutoCoder (33B) and AutoCoder-S (6.7B). As shown in Figure 1, AutoCoder demonstrates higher accuracy. In the HumanEval Base Test, we compared our results with several models featured on the EvalPlus Leaderboard as of September 2024 evalplusleaderboard (2024). The performance of Claude 3.5 Sonnet Anthropic (2024) was obtained from its official website, while the GPT-4o 2024-08-06 results were self-implemented. Remarkably, AutoCoder's Pass@1 even outperforms some of the top-ranked models, including GPT-4 Turbo 2024-04-09 and GPT-4o 2024-08-06.

Moreover, as illustrated in Figure 2, AutoCoder boasts a more versatile Code Interpreter function compared to GPT-4o. The Code Interpreter is an external program execution environment that large models utilize to execute the code they deem necessary. While GPT-4o and GPT-4 Turbo can identify the code that needs to be executed, they fail to provide the Code Interpreter with the necessary instructions to install external packages required by the programs. This limitation significantly restricts the capabilities of the Code Interpreter. In contrast, AutoCoder can correctly supply the Code Interpreter with the appropriate external package installation instructions, thereby enabling it to execute a wide variety of code. **As far as we know, as of September 2024, AutoCoder is the only model that supports automatically installing external packages in the Code Interpreter.**

To comprehensively evaluate the capabilities of AutoCoder, we tested it on several datasets: HumanEval Chen et al. (2021), HumanEval+ Liu et al. (2024), MBPP Austin et al. (2021), MBPP+ Liu et al. (2024), MultiPL-E Cassano et al. (2022), DS-1000 Lai et al. (2023) and LiveCodeBench Jain et al. (2024) . To analyze the contribution of different components to AutoCoder's performance, we compared it to its base model, Deepseek-Coder Guo et al. (2024). The performance and detailed experimental procedures can be found in Section 5.

Overall, our contributions are summarized as follows:

**We propose AIEV-INSTRUCT**, a novel method for creating high-quality large code datasets. It simulates programmers writing code and conducting unit tests through agent interactions, ensuring annotation accuracy with an external code executor. It includes a *Teaching Stage* and a *Self-Learning Stage*, reducing reliance on expensive closed-source models during the annotation process.

**We introduce AutoCoder,** a code LLM trained using AIEV-INSTRUCT that excels in code-related tasks. It outperforms top models like GPT-4 Turbo and GPT-4o on the HumanEval benchmark.

**We enhances the functionality of the current code interpreters.** AutoCoder can provide the code interpreter with the necessary instructions to install external packages, extending the applicability of the code interpreter beyond built-in packages.

## 2 RELATED WORK

**Large Language Models for Code.** Recently, LLMs have shown remarkable abilities in understanding and generating code Kazemitabaar et al. (2023). Trained on extensive datasets covering various programming languages and tasks, these models excel in code completion, bug fixing, and code synthesis Jin et al. (2023). Closed-source models like OpenAI's GPT-4 OpenAI (2024), Claude.ai's Claude AnthropicAIteam (2024), and Google's Gemini deepmindteam (2024) series have demonstrated superior performance on code tasks. Meanwhile, open-source models specialized for code, such as DeepSeek-Coder deepseekteam (2024), CodeQwen Qwen (2024), Magicoder Wei et al. (2023), OpenCodeInterpreter Zheng et al. (2024), and WizardCoder Luo et al. (2023), are also emerging. Generally, closed-source models outperform open-source ones due to their larger parameter sizes and broader knowledge base.

**Code LLMs Instruction Tuning.** After pre-training large models, we use instruction tuning to optimize them Gao et al. (2020), enhancing their ability to understand and execute specific instructions Chang et al. (2024). A major challenge in Instruction Tuning for Code LLMs is the lack of high-quality instruction datasets for code Rao (2024). Code tasks, such as Text-Code and Code-Code translation, are difficult and time-consuming to annotate manually. OpenAI used human annotators to label various tasks and train InstructGPT Ouyang et al. (2022), but they noted that annotating code tasks is prohibitively expensive for large-scale datasets. Since the advent of GPT-4, an increasing number of researchers have leveraged GPT-4 for code annotation to create high-quality instruction tuning datasets. Currently, there are three primary methods: SELF-INSTRUCT Wang et al. (2022), EVOL-INSTRUCT Luo et al. (2023), and OSS-INSTRUCT Wei et al. (2023). SELF-INSTRUCT boosts LLMs' instruction-following skills by using strong teacher models to generate synthetic coding instructions for fine-tuning weaker student models. EVOL-INSTRUCT iteratively enhances LLMs' coding abilities by increasing the complexity of seed code instructions. OSS-INSTRUCT creates diverse coding problems inspired by open-source code snippets. These methods distill the expertise of powerful teacher models like GPT-4 to guide and improve smaller models.

## 3 AIEV-INSTRUCT

### 3.1 OVERALL ARCHITECTURE

Figure 3 illustrates the overall architecture of AIEV-INSTRUCT, divided into two stages: the *Teaching Stage* and the *Self-Learning Stage*. In the *Teaching Stage*, the model learns primarily by distilling knowledge from a teacher model. In the *Self-Learning Stage*, it learns autonomously.

In the *Teaching Stage*, we obtain open-source code snippets and use GPT-4 Turbo as the teacher model to supplement and correct them. The process consists of four main steps. **In I : Initialization,** we initialize the necessary components. GPT-4 Turbo is assigned two roles: *questioner* and *programmer*. It can ensure the generated data is diverse, resulting in a more uniform probability distribution rather than converging to a specific dialogue template. The dialogue messages are initialized as an empty list, which will be used throughout the process to store data. Eventually, this list will contain multiple rounds of dialogue, and the entire conversation will be added as a single data entry to our final dataset. Additionally, we need to initialize a Docker container as our Code Interpreter. This container is responsible for installing the required external packages and executing the code that needs verification throughout the process. **In II : Propose the question,** we first utilize GPT-4 Turbo

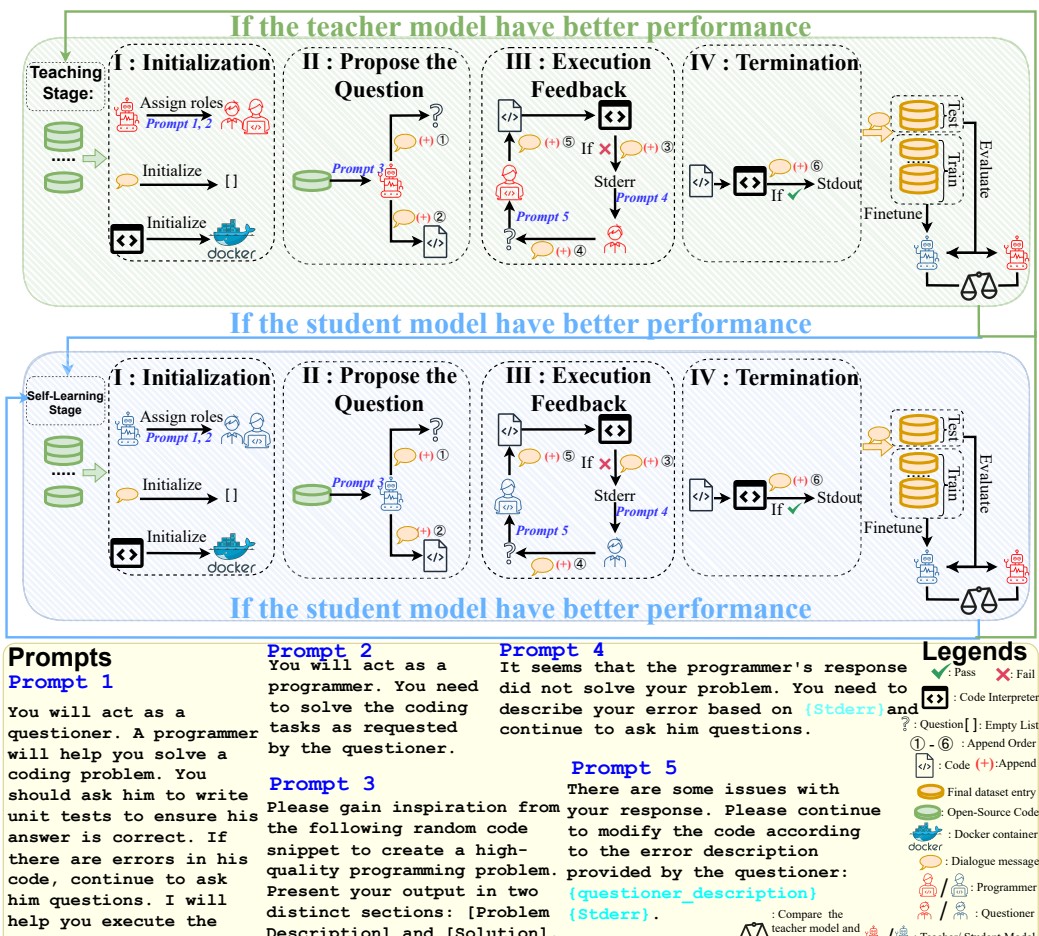

Figure 3: The overall architecture of the AIEV-INSTRUCT.

to execute OSS-Instruct Wei et al. (2023), designing a problem description and a specific solution that includes the code snippet based on the open-source code fragment. The difference here is that we require GPT-4 Turbo to provide some **Unit Tests**. These **Unit Tests** further ensure the accuracy of the code in our dataset. The dialogue messages initialized in the previous step are sequentially appended with the problem description (①), the solution and the unit tests (②). **In III: Execution Feedback:**, we use multiple rounds of execution feedback to check the generated code, thereby improving the quality of the dataset. First, we input the code snippet generated in the second step into the Code Interpreter. If an execution error occurs, the dialogue messages append the detailed Stderr output (③). Meanwhile, this Stderr information is provided to the *questioner*, who will generate a natural language description based on the Stderr. This natural language description is also appended to the dialogue messages (④). Next, both the natural language description and the Stderr are provided as new questions to the *programmer*, who will continue to modify the code. The dialogue messages will append the new code it generates (⑤) and repeat this process. **In IV: Termination**, we also use the Code Interpreter to run the code generated by the *programmer*. If the program executes successfully, the Stdout is appended to the dialogue messages (⑥). This completes the analysis of one data entry.

After analyzing every 2000 data entries, we split the new data into a test set and a training set in a 1:9 ratio. The training set is used to train the student model (AutoCoder). After training, we use the test set to evaluate both the teacher model and the student model. Upon completion of the evaluation, we compare the Pass@1 of the two models. If the teacher model performs better, we continue executing the *Teaching Stage*. If the student model performs better, we move to the *Self-Learning Stage*. The difference between the *Self-Learning Stage* and the *Teaching Stage* is that in the *Self-Learning Stage*, we replace the original teacher model with the student model. The student model itself is assigned as the *questioner* and *programmer* , and it completes the entire execution feedback process.

## 3.2 DATASET ANALYSIS

**Dataset Generation.** To prevent **data contamination** in test sets from resulting in overly high performance on certain benchmark datasets (such as HumanEval), we used code from two datasets that had already undergone contamination detection: *Magicoder-Evol-Instruct* and *Magicoder-OSS-Instruct* Wei et al. (2023). We collected a total of 186K original code entries from these two datasets. After de-duplication, we input these data into our AIEV-Instruct pipeline to generate the dataset. We set the maximum number of execution feedback iterations in AIEV-Instruct to 7. If the generated code fails to execute successfully and pass all unit tests after 7 attempts, that data point is discarded. The `gpt-4-turbo-2024-04-09` is used as the teacher model. Sample demonstrations of some data points are provided in the Appendix D.

**Dataset Comparision.** We compared our dataset *AutoCoder-AIEV-Instruct* with several current large code instruction datasets. The comparison results are shown in Figure 4. The dataset *AutoCoder-AIEV-Instruct* contains 169K data samples, totaling 241K rounds of dialogue. Among these, 150K rounds are contributed by multi-round dialogue data samples. Besides including the main function, it also encompasses subsequent package installations, code execution errors, or results, as well as various error analyses. Compared to the original *Magicoder-Evol-Instruct* and *Magicoder-OSS-Instruct*, it adds unit tests, which further enhances the accuracy of code-related tasks. Additionally, compared to *Code-Feedback* Zheng et al. (2024), it includes more execution feedback results, reducing the multi-round dialogues for code block concatenation and enhancing the coherence of the context.

**Dataset Decontamination.** Similar to the data processing method used by StarCoder Li et al. (2023b), we also performed decontamination for *AutoCoder-AIEV-Instruct*. Specifically, we tested each code snippet from HumanEval, MBPP, DS-1000, and MultiPL-E against every code snippet in *AutoCoder-AIEV-Instruct* using Levenshtein distance. If the similarity exceeded 90%, the data entry was removed. Through this process, we excluded a total of 113 data entries.

**Dataset Accuracy Theoretical Analysis.** Although our main conclusions are derived from the experiments, we provide some theoretical analysis in the Appendix B to explain why datasets generated using the AIEV-INSTRUCT method achieve higher accuracy compared to previous OSS-INSTRUCT and EVOL-INSTRUCT methods. Specifically, $\mathcal{A}_{\text{Evol}} < \mathcal{A}_{\text{OSS}} < \mathcal{A}_{\text{AIEV}}$.

## 4 AUTOCODER

### 4.1 CODE INTERPRETER

Code Interpreter assists the model in debugging and executing code, which is essential for fully automating complex coding, scientific computations, and related tasks. Building a code interpreter requires the model to accurately identify the code blocks it needs to run. Currently, only a few models, like GPT-4 Turbo and InternLM-Chat Cai et al. (2024), support code interpreters. However, a significant limitation of these interpreters is that they operate in a closed environment and cannot interact with external systems, preventing them from executing code that requires external package installations. AutoCoder addresses this issue by enabling the execution of `bash` commands to install necessary packages. This capability is achieved by teaching the model to run `bash` commands when appropriate. To facilitate this, we need to perform some post-processing on the *AutoCoder-AIEV-Instruct* dataset.

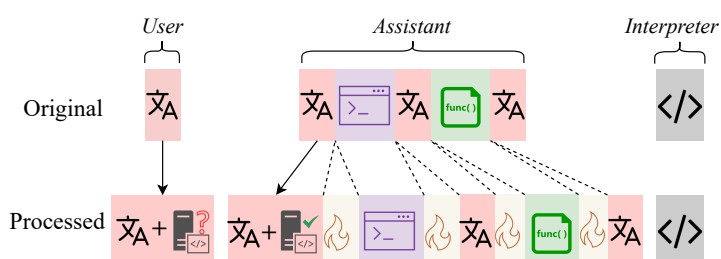

Figure 5: *AutoCoder-AIEV-Instruct* dataset post-processing. ✗A:Nature language; :Code execution request from the User; :Code execution request response from the Assistant; :Bash command; :Code block; :Special token; </>:Execution result.

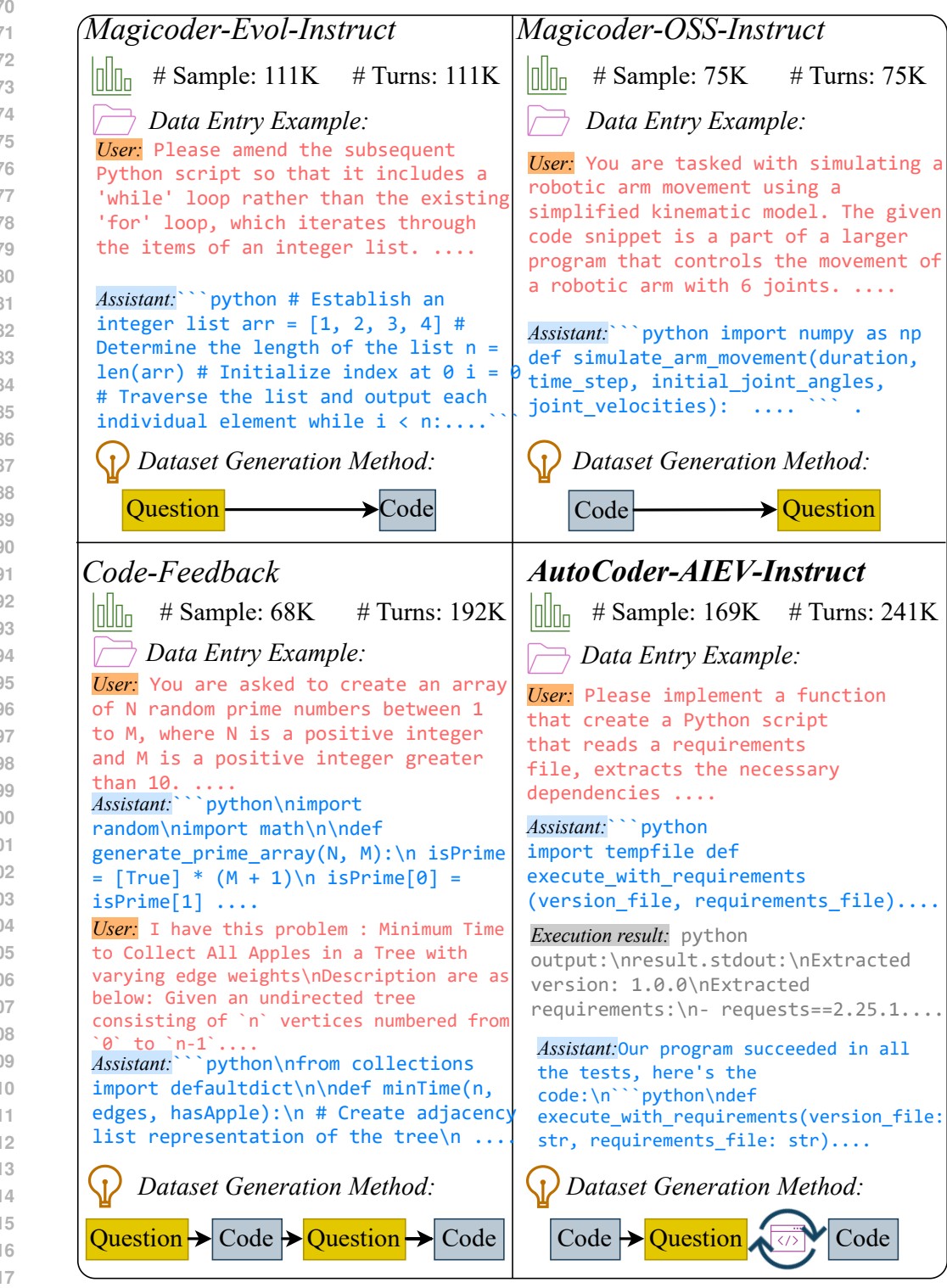

Figure 4: The comparison between the AutoCoder-AIEV-Instruct and other large code datasets.

As shown in Figure 5, for a simple single execution feedback example, the original data entry contains three parts: natural language from the **User**; natural language + `bash` command + natural language + code block + natural language from the **Assistant**; execution result from the **code interpreter**.

In the post-processing stage, we mix the natural language of the Code execution request into the User's natural language, enabling the model to correctly learn when to execute the code. Then, we mix the code execution request response into the Assistant's response, so it can generate coherent answers. Finally, we add special tokens before and after the `bash` commands and code blocks in the Assistant's original response, allowing the model to learn to correctly identify the `bash` commands and code blocks that need to be executed. Sample demonstrations of data points after post-processing are provided in the Appendix D.

## 4.2 TRAINING DETAILS

We fine-tuned two base models, Deepseek-Coder 6.7B and 33B, using the *AutoCoder-AIEV-Instruct* dataset to obtain our AutoCoder 33B and AutoCoder-S 6.7B. We utilized the `AutoTokenizer` package from the `transformer` library to add four special tokens to these models to enable the Code Interpreter feature for AutoCoder. For hardware, we used 10 nodes with a total of 40 80GB A100 GPUs on a Simple Linux Utility for Resource Management (SLURM) cluster. The NVIDIA Collective Communications Library (NCCL) handled communication between GPUs. In terms of training parameters, we used the ZeRO-Stage 3 feature from the `deepspeed` library to partition model parameters, with a batch size of 8 per GPU, a gradient accumulation step of 4, a learning rate of 5e-5, and bf16 as the parameter type. The max sequence length was set to 5120 and the total epochs was set to 2. We adopted a full-parameter tuning approach to train the model.

## 5 EXPERIMENT

We tested AutoCoder's capabilities in Python text-to-code generation, multilingual code generation, as well as code generation for data science questions and challenging coding problems. To ensure a fair comparison with other models and reduce experimental randomness, we disabled AutoCoder's external code interpreter during the tests. Due to the large number of models, for each dataset, We only selected certain models from the corresponding leaderboard for comparison based on: ① Well-known closed-source or large-parameter models, and ② Models with a similar number of parameters to AutoCoder. To facilitate reading, for Tables 1, 2, and 3, we used red, blue, and brown to label the data points ranked $1st$, $2nd$, and $3rd$ in each dataset, respectively. In Table 4, we highlighted the best-performing models among those with similar parameter sizes in **bold**. The specific parameters of the model during inference are presented in the Appendix A. In addition to AutoCoder and AutoCoder-S, we also fine-tuned Codellama-7B and CodeQwen1.5-7B using AutoCoder-AIEV-INSTRUCT dataset and evaluated their performance. Their experimental results are provided in the Appendix C.

## 5.1 PYTHON TEXT TO CODE GENERATION

In Table 1, we evaluated AutoCoder using two of the most commonly used code generation benchmarks: HumanEval Chen et al. (2021) and MBPP Austin et al. (2021). HumanEval is widely used to test various state-of-the-art closed-source models, such as GPT-4o OpenAI (2024), Claude 3.5 Sonnet Anthropic (2024), Gemini Ultra 1.0 DeepMind (2024), and Llama3.1 405b AI (2024a). It contains 164 code generation problems. Compared to HumanEval, MBPP has more test data, with a total of 378 test cases. Additionally, to prevent errors due to the insufficient number of test cases for each code problem in the original benchmarks, HumanEval+ and MBPP+ Liu et al. (2024) have added more test cases to the original datasets.

Experimental results demonstrate that AutoCoder-33B achieved a Pass@1 score of $90.9\%$ on the HumanEval benchmark, ranking just below Claude 3.5 Sonnet as of September 2024 when compared to other state-of-the-art code LLMs. On HumanEval+, it achieved a Pass@1 score of $78\%$, significantly outperforming models with fewer than 70B parameters. In the MBPP and MBPP+ tests, AutoCoder-33B achieved Pass@1 scores of $82.5\%$ and $70.6\%$, respectively, leading among models with 33B parameters or fewer. Additionally, despite having only 6.7B parameters, AutoCoder-S also delivered impressive results, achieving $78.7\%$ on HumanEval and $72\%$ on HumanEval+. For the MBPP and MBPP+ benchmarks, it scored $79.4\%$ and $69.8\%$, respectively. Remarkably, on MBPP+, its performance even surpassed some models in the 70B parameter range.

Table 1: Comparison with the current SOTA code large language models on HumanEval(+) and MBPP(+). The results for GPT-4o, Llama3.1-Instruct, Claude 3.5 Sonnet, DeepSeek-Coder-V2-Instruct, Qwen2.5-Instruct, and Codestral are sourced from their official websites or technical reports OpenAI (2024); AI (2024a); Anthropic (2024); Zhu et al. (2024); AI (2024c;b), while the remaining results are obtained from the EvalPlus leaderboard evalplusleaderboard (2024).

| Model | Size | Benchmark (Pass@1 %) | | | |
|---|---|---|---|---|---|
| | | HumanEval | HumanEval+ | MBPP | MBPP+ |
| GPT-4o `2024-08-06` | 🔒 | **90.2** | - | - | - |
| GPT-4 Turbo `2024-04-09` | 🔒 | **90.2** | 86.6 | 85.7 | 73.3 |
| Claude 3.5 Sonnet | 🔒 | **92.0** | - | 90.5 | - |
| Llama3.1-Instruct | $405B$ | 89.0 | - | 88.6 | - |
| DeepSeek-Coder-V2-Instruct | $236B$ | **90.2** | - | - | 76.2 |
| Qwen2.5-Instruct | $72B$ | 86.6 | - | 88.2 | - |
| OpenCodeInterpreter-CL | $70B$ | 76.2 | 70.7 | 73.0 | 61.9 |
| CodeLlama-Instruct | $70B$ | 72.0 | 65.2 | 75.4 | 61.7 |
| DeepSeek-Coder-instruct | $33B$ | 81.1 | 75.0 | 80.4 | 70.1 |
| WizardCoder-V1.1 | $33B$ | 79.9 | 73.2 | - | - |
| OpenCodeInterpreter-DS | $33B$ | 79.3 | 73.8 | 80.2 | 68.5 |
| speechless-codellama-v2.0 | $34B$ | 77.4 | 72.0 | 73.8 | 61.4 |
| Codestral | $22B$ | 81.1 | - | 78.2 | - |
| OpenCodeInterpreter-CL | $13B$ | 77.4 | 73.8 | 70.7 | 59.2 |
| starchat2-v0.1 | $15B$ | 73.8 | 71.3 | 74.9 | 64.6 |
| starcoder2-instruct-v0.1 | $15B$ | 67.7 | 60.4 | 78.0 | 65.1 |
| OpenCodeInterpreter-DS | $6.7B$ | 77.4 | 72.0 | 76.5 | 66.4 |
| Artigenz-Coder-DS | $6.7B$ | 75.6 | 72.6 | 80.7 | 69.6 |
| DeepSeek-Coder-instruct | $6.7B$ | 74.4 | 71.3 | 74.9 | 65.6 |
| AutoCoder | $33B$ | 90.9 | 78.0 | 82.5 | 70.6 |
| AutoCoder-S | $6.7B$ | 78.7 | 72.0 | 79.4 | 69.8 |

## 5.2 MULTILINGUAL CODE GENERATION

In Table 2, we tested AutoCoder's capabilities in multilingual code generation, we used MultiPL-E benchmark Cassano et al. (2022) to evaluate its performance in six additional commonly used languages. Since MultiPL-E's official library does not support testing closed-source models, we ensured consistent experimental conditions by comparing only with well-known open-source models.

The experimental results show that AutoCoder performed exceptionally well in Java, C++, and Rust, achieving 61.4%, 68.9%, and 60.8% Pass@1 respectively. In the other three languages, its performance was only surpassed by a few models such as Qwen2.5-Instruct-72B and Llama-3.1-Instruct-70B. This demonstrates AutoCoder's robust capabilities in multilingual code generation.

## 5.3 CODE GENERATION FOR DATA SCIENCE

In Table 3, we tested AutoCoder's ability to generate code to solve data science problems using the DS-1000 dataset Lai et al. (2023). It contains 1000 questions that require the use of seven commonly used Python data science libraries. We tested all the models using the *completion* mode in DS-1000.

The result shows that the AutoCoder's Pass@1 on Matplotlib-related questions even surpassed that of GPT-4 Turbo. Overall, AutoCoder achieves a Pass@1 rate of 47.2%, which is higher than other models with the same parameter count and even surpasses some closed-source models. This demonstrates AutoCoder's excellent capability to generate code for data science problems.

## 5.4 PERFORMANCE ON MORE CHALLENGING CODE PROBLEMS

In Table 4, we tested AutoCoder on more challenging code problems using the LiveCodeBench dataset Jain et al. (2024) (`2024-09`). The LiveCodeBench dataset collects new problems over time from contests across three competition platforms, namely LeetCode, AtCoder, and CodeForces. It is regularly maintained and updated over time to ensure the dataset remains uncontaminated.

Table 2: Performance (Pass@1 %) of AutoCoder on the MultiPL-E benchmark.

| Model | Size | Programming Language | | | | | |
|---|---|---|---|---|---|---|---|
| | | Java | JavaScript | C++ | PHP | Swift | Rust |
| Qwen2.5-Instruct | 72B | 68.3 | 79.2 | 67.6 | 77.3 | 59.6 | 55.4 |
| Llama-3.1-Instruct | 70B | 60.3 | 73.1 | 65.2 | 67.4 | 52.4 | 57.9 |
| OpenCodeInterpreter-CL | 70B | 52.3 | 62.9 | 64.2 | 59.8 | 48.7 | 50.4 |
| Wizard-CL | 34B | 44.9 | 55.3 | 47.2 | 47.2 | 44.3 | 46.2 |
| CodeLLAMA-Instruct | 34B | 41.5 | 45.9 | 41.5 | 37 | 37.6 | 39.3 |
| Deepseek-Coder-Instruct | 33B | 53.8 | 67.7 | 63.3 | 54.7 | 51.3 | 54.4 |
| OpenCodeInterpreter-DS | 33B | 60.1 | 69.6 | 67.1 | 59.6 | 54.4 | 60.2 |
| StarCoder-Base | 15B | 28.5 | 31.7 | 30.6 | 26.8 | 16.7 | 24.5 |
| StarCoder | 15B | 30.2 | 30.8 | 31.6 | 26.1 | 22.7 | 21.8 |
| WizardCoder-SC | 15B | 35.8 | 41.9 | 39.0 | 39.3 | 33.7 | 27.1 |
| CodeLLAMA | 7B | 29.3 | 31.7 | 27.0 | 25.1 | 25.6 | 25.5 |
| Magicoder-CL | 7B | 36.4 | 45.9 | 36.5 | 39.5 | 33.4 | 30.6 |
| MagicoderS-CL | 7B | 42.9 | 57.5 | 44.4 | 47.6 | 44.1 | 40.3 |
| AutoCoder-S | 6.7B | 55.7 | 65.2 | 62.7 | 59.6 | 41.1 | 50.6 |
| AutoCoder | 33B | 61.4 | 68.9 | 68.9 | 63.4 | 53.8 | 60.8 |

Table 3: Performance (Pass@1 %) of AutoCoder on the DS-1000 dataset. plt: Matplotlib, np: NumPy , Pd: Pandas, Py: PyTorch, Scp: Scipy, Sk: Sklearn , TF: TensorFlow. The result of GPT-4 Turbo `2024-04-09`, GPT-3.5 Turbo `0125` and Codex-002 are from the Offical Github of DS-1000 AI (2023). * DS-Coder-Instruct: Deepseek-Coder-Instruct; OC-DS: OpenCodeInterpreter-DS.

| Model | Size | 155 | 220 | 291 | 68 | 106 | 115 | 45 | 1000 |
|---|---|---|---|---|---|---|---|---|---|
| | | plt | np | Pd | Py | Scp | Sk | TF | Overall |
| GPT-4 Turbo | 🔒 | 72.3 | 61.8 | 42.3 | 50.0 | 50.0 | 50.4 | 53.3 | 53.9 |
| GPT-3.5 Turbo | 🔒 | 65.8 | 32.7 | 30.2 | 36.8 | 39.6 | 40 | 42.2 | 39.4 |
| Codex-002 | 🔒 | 57 | 43.1 | 26.5 | 41.8 | 31.8 | 44.8 | 39.3 | 39.2 |
| DS-Coder-Instruct * | 33B | 61.3 | 50.0 | 30.9 | 35.3 | 36.8 | 45.2 | 40.0 | 42.8 |
| OC-DS * | 33B | 39.4 | 57.7 | 28.2 | 47.1 | 40.6 | 49.6 | 42.2 | 42.1 |
| CodeGen-Mono | 16B | 31.7 | 10.9 | 3.40 | 7.00 | 9.00 | 10.8 | 15.2 | 11.7 |
| StarCoder | 15B | 51.7 | 29.7 | 11.4 | 21.4 | 20.2 | 29.5 | 24.5 | 26.0 |
| WizardCoder-SC | 15B | 55.2 | 33.6 | 16.7 | 26.2 | 24.2 | 24.9 | 26.7 | 29.2 |
| CodeLlama-Python | 7B | 55.3 | 34.5 | 16.4 | 19.9 | 22.3 | 17.6 | 28.5 | 28.0 |
| WizardCoder-CL | 7B | 53.5 | 34.4 | 15.2 | 25.7 | 21.0 | 24.5 | 28.9 | 28.4 |
| Magicoder-CL | 7B | 54.6 | 34.8 | 19.0 | 24.7 | 25.0 | 22.6 | 28.9 | 29.9 |
| MagicoderS-CL | 7B | 55.9 | 40.6 | 28.4 | 40.4 | 28.8 | 35.8 | 37.6 | 37.5 |
| InCoder | 6.7B | 28.3 | 4.4 | 3.1 | 4.40 | 2.80 | 2.80 | 3.80 | 7.40 |
| AutoCoder-S | 6.7B | 52.9 | 38.2 | 31.6 | 30.9 | 31.1 | 39.1 | 31.1 | 37.1 |
| AutoCoder | 33B | 72.9 | 52.7 | 36.1 | 26.5 | 45.3 | 46.1 | 42.2 | 47.2 |

The experimental results show that at the 33B parameter scale, AutoCoder outperforms other models of the same scale, particularly on Medium difficulty problems, surpassing DeepSeek-Coder-Instruct-33B by 3.7 percentage points.

## 5.5 IMPACT OF DIFFERENT COMPONENTS

To further understand the impact of different components in the dataset on the final model performance, we compared the following: the Base model (DeepSeek-Coder-Base) itself, the model fine-tuned on a single-turn dialogue dataset, the model fine-tuned on a multi-turn dialogue dataset, and the model fine-tuned on a multi-turn dialogue dataset that includes code execution feedback with unit tests (AutoCoder). We conducted experiments using the HumanEval, MBPP, and DS-1000 datasets.

As shown in Figure 6, across all three datasets, AutoCoder demonstrated superior performance compared to other models, especially on the DS-1000 dataset. AutoCoder-S (6.7B) outperformed the model trained only on the multi-turn dialogue dataset by 3.4 percentage points, while AutoCoder

Table 4: Performance (Pass@1 %) of AutoCoder on the LiveCodeBench dataset.

| Model | Size | W/Wo CoT | Total Pass@1 | Easy Pass@1 | Medium Pass@1 | Hard Pass@1 |
|---|---|---|---|---|---|---|
| o1-Mini | 🔒 | ✓ | **73.1** | **94.3** | **76.6** | **38.8** |
| o1-Preview | 🔒 | ✓ | 57.3 | 91.2 | 54.9 | 14.7 |
| Claude-3.5-Sonnet | 🔒 | ✗ | 51.3 | 87.2 | 45.3 | 11 |
| GPT-4o 2024-08-06 | 🔒 | ✗ | 46.1 | 89.5 | 34.9 | 3.5 |
| GPT-4 Turbo 2024-04-09 | 🔒 | ✗ | 44.2 | 85 | 32.6 | 5.8 |
| DeepSeekCoder-V2 | 236B | ✗ | 41.9 | 79.9 | 32 | 4.9 |
| Qwen2-72B-Instruct | 72B | ✗ | **30.1** | **65.7** | **16.3** | **2.2** |
| LLaMA3-70b-Ins | 70B | ✗ | 27.4 | 59.4 | 15.6 | 1.3 |
| Qwen2-Base | 72B | ✗ | 21.2 | 50.6 | 8.1 | 0.7 |
| AutoCoder | 33B | ✗ | **25.4** | **56.6** | **12.3** | **0.9** |
| DeepSeek-Coder-Instruct | 33B | ✗ | 23.4 | 56.1 | 8.6 | 0.9 |
| Command-R+ | 35B | ✗ | 21.3 | 53.1 | 6.1 | 0.5 |
| OpenCodeInterpreter-DS | 33B | ✗ | 20.6 | 52.2 | 5.4 | 0 |
| Phind-34B-V2 | 34B | ✗ | 19.9 | 51.6 | 4.3 | 0.1 |
| StarCoder2-15B | 15B | ✗ | **14.6** | **37.6** | **3.5** | 0 |
| CodeLlama-13B-Instruct | 13B | ✗ | 13.4 | 35 | 2.4 | **0.3** |
| CodeLlama-13B-Base | 13B | ✗ | 8.5 | 23.2 | 0.9 | 0 |

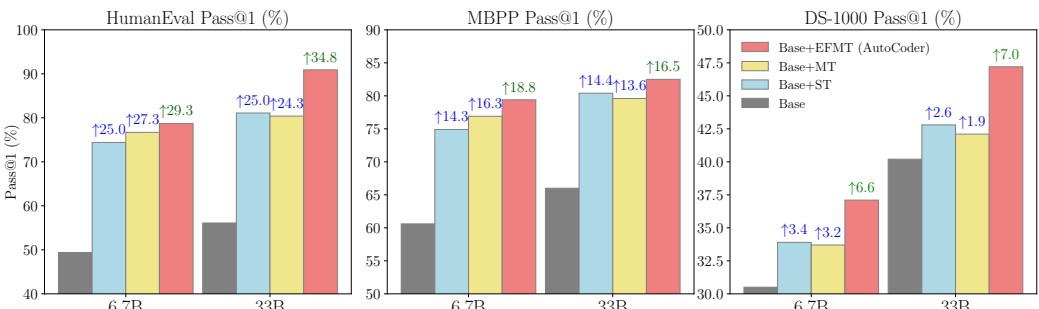

Figure 6: Comparison of AutoCoder with other models sharing the same base model. Base: Base model; Base + ST: Base model fine-tuned on a single-turn dialogue dataset; Base + MT: Base model fine-tuned on a multi-turn dialogue dataset; Base + EFMT: Base model fine-tuned on a multi-turn dialogue dataset that includes code execution feedback with unit tests (AutoCoder).

outperformed it by 5.1 percentage points. This proves that fine-tuning the model on a dataset generated by AIEV-INSTRUCT can effectively enhance its code capabilities.

## 6 CONCLUSION

We propose AIEV-INSTRUCT, a novel method for creating high-quality code instruction datasets. It simulates programmers writing code and conducting unit tests through agent interactions, ensuring accuracy with execution validation. It includes both a *teaching stage* and a *self-learning stage*, reducing reliance on expensive closed-source models during the annotation process. Using the dataset generated with AIEV-INSTRUCT, we trained the AutoCoder code LLM. It exhibits excellent performance and surpass the current top models, GPT-4 Turbo and GPT-4o on the HumanEval benchmark. Furthermore, AutoCoder extends the functionality of previous code interpreters by allowing them to automatically install external packages, thus extending the applicability of the code interpreter. Overall, our work provides the community with excellent open-source code large language models and offers new insights for generating high-quality large code instruction dataset.

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

## A  EXPERIMENT

Our experiments followed the standard testing procedures from the official repositories of each dataset, and the specific parameters for model inference are shown in Table 5.

Table 5: The parameters for performing model inference in each experiment. *: Default value from their official repository. DS: DeepSeek-Coder prompting style.

| Experiment | Official Repo | Temperature | TopP | TopK | Prompting Method | Prompt |
|---|---|---|---|---|---|---|
| Section 5.1 | evalplus | 0.0 | - | - | 0-shot | DS |
| Section 5.2 | multiple | 0.2 | 0.95 | - | * | * |
| Section 5.3 | ds1000 | * | * | * | * | * |
| Section 5.4 | livecodebench | * | * | * | * | DS |
| Section 5.5 | - | 0.0 | - | - | 0-shot | DS |

## B  DATASET ACCURACY THEORETICAL ANALYSIS

In this analysis, we compare the theoretical maximum accuracies of different datasets (EVOL-INSTRUCT, OSS-INSTRUCT, and AIEV-INSTRUCT) to understand how various factors, such as problem generation, code alignment, and iterative validation, affect the overall accuracy of code generation models. The following assumptions is introduced.

**Assumption 1. (Teacher Model Accuracy for Code Generation)** The theoretical maximum accuracy of the EVOL-INSTRUCT dataset should closely match the teacher model's accuracy in generating correct code $c$ for given problems $p$. Therefore, $\mathcal{A}_{\text{Evol}} \approx \mathcal{P}(c \mid p)$.

**Assumption 2. (Teacher Model Accuracy for Problem Generation)** The theoretical maximum accuracy of the OSS-INSTRUCT dataset should closely match the teacher model's accuracy in analyzing and interpreting open-source code $c$. Therefore, $\mathcal{A}_{\text{OSS}} \approx \mathcal{P}(p \mid c)$.

**Assumption 3. (Rarity of Code)** The number of valid code solutions $c$ for a problem $p$ is smaller than the number of possible problem descriptions. Thus, $\mathcal{P}(c) < \mathcal{P}(p)$.

**Assumption 4. (Improved Alignment through Unit Tests)** Adding unit tests to the original code improves the alignment of problem descriptions and code. Therefore, we assume $\mathcal{P}(p \mid c) < \mathcal{P}(p^* \mid c^*)$, where $p^*$ and $c^*$ represent the new problem description and code after adding unit tests.

**Assumption 5. (Iterative Validation Increases Correctness)** The probability of correctness improves with each iteration during the iterative validation and correction process. If the probability of error in each iteration is $1 - \mathcal{P}(p^* \mid c^*)$, then the probability of correctness after $n$ iterations is $\mathcal{A}_{\text{AIEV}} \approx 1 - (1 - \mathcal{P}(p^* \mid c^*))^n$.

*Remark* 1. All assumptions above are mild. Assumption 1 is necessary to establish that the theoretical accuracy of the EVOL-INSTRUCT dataset is fundamentally linked to the teacher model's capacity to generate correct code for a given problem. Assumption 2 reflects that the OSS-INSTRUCT dataset's accuracy depends on the teacher model's ability to interpret and understand the open-source code, a natural requirement for problem generation. Assumption 3 acknowledges the common observation that there are fewer valid code solutions than possible problem descriptions, making code inherently rarer. This assumption is essential for applying Bayes' theorem in the analysis. Assumption 4 states that adding unit tests to the original code enhances the alignment between the code and its corresponding problem description, which is a well-accepted practice in software engineering to ensure code correctness. Finally, Assumption 5 posits that iterative validation and correction increase the probability of achieving a correct solution. This is a common concept in optimization processes, where each iteration helps refine and improve the overall accuracy.

Based on these assumptions, the following theorem is derived.

**Theorem 1. (*Relative Accuracy of Datasets*)** *Given the assumptions 1 - 5, the accuracy of* AUTOCODER-AIEV-INSTRUCT *is higher than that of* MAGICODER-OSS-INSTRUCT*, and the accuracy of* MAGICODER-OSS-INSTRUCT *is higher than* MAGICODER-EVOL-INSTRUCT*, that is,*

$$\mathcal{A}_{\text{Evol}} < \mathcal{A}_{\text{OSS}} < \mathcal{A}_{\text{AIEV}}.$$

*Proof.* From Assumption 1, we have $\mathcal{A}_{\text{Evol}} \approx \mathcal{P}(c \mid p)$. By applying Bayes' theorem, one can get $\mathcal{P}(c \mid p) = \frac{\mathcal{P}(p|c) \cdot \mathcal{P}(c)}{\mathcal{P}(p)}$. Based on the Assumption 3, it follows that $\mathcal{A}_{\text{Evol}} \approx \frac{\mathcal{P}(p|c) \cdot \mathcal{P}(c)}{\mathcal{P}(p)} < \mathcal{P}(p \mid c)$.

From Assumption 2, the accuracy of the OSS-INSTRUCT dataset is given by $\mathcal{A}_{\text{OSS}} \approx \mathcal{P}(p \mid c)$.

Next, by Assumption 4, adding unit tests improves the alignment, so $\mathcal{P}(p \mid c) < \mathcal{P}(p^* \mid c^*)$.

By Assumption 5, the probability of correctness after $n$ iterations in the iterative validation process is $\mathcal{A}_{\text{AIEV}} \approx 1 - (1 - \mathcal{P}(p^* \mid c^*))^n$. Since $\mathcal{P}(p^* \mid c^*) > \mathcal{P}(p \mid c)$ and $n > 1$, one can get $1 - (1 - \mathcal{P}(p^* \mid c^*))^n > 1 - (1 - \mathcal{P}(p \mid c))^n$.

Thus, $\mathcal{A}_{\text{AIEV}} > \mathcal{A}_{\text{OSS}}$. Combining these results, one can get

$$\mathcal{A}_{\text{Evol}} \approx \frac{\mathcal{P}(p \mid c) \cdot \mathcal{P}(c)}{\mathcal{P}(p)} < \mathcal{P}(p \mid c) \approx \mathcal{A}_{\text{OSS}} < 1 - (1 - \mathcal{P}(p \mid c))^n < \mathcal{A}_{\text{AIEV}},$$

that is, $\mathcal{A}_{\text{Evol}} < \mathcal{A}_{\text{OSS}} < \mathcal{A}_{\text{AIEV}}$. This completes the proof. $\square$

## C    PERFORMANCE OF ADDITIONAL MODELS

We fine-tuned Codellama-7B and CodeQwen1.5-7B using AutoCoder-AIEV-INSTRUCT dataset and evaluated their performance. The results are shown in Table 6. The inference parameters of the models are consistent with those in Section 5.1. The results show a significant performance improvement for the older CodeLlama-7B model when fine-tuned with the AutoCoder-AIEV-INSTRUCT dataset. Similarly, for the more recent CodeQwen-1.5-7B model, the AutoCoder-AIEV-INSTRUCT dataset also led to further performance enhancement.

Table 6: Comparison of the performance between AutoCoder-CL-7B and AutoCoder-QW-7B with their respective base models. CL: CodeLlama; QW: Qwen.

| Model | HumanEval | HumanEval+ | MBPP | MBPP+ |
|---|---|---|---|---|
| CodeLlama-7B | 37.8 | 34.1 | 57.6 | 45.4 |
| AutoCoder-CL-7B | 72.7 | 69.3 | 69.8 | 60.4 |
| CodeQwen1.5-7B | 83.5 | 78.7 | 79.4 | 69 |
| AutoCoder-QW-7B | 86.4 | 80.2 | 80.4 | 69.7 |

## D    MORE DATA SAMPLES

Below we provide several examples to explain what the data look like. In these examples, we use **bold** to indicate the **special token**. We use ***bold and italic*** for explanatory text that is **NOT** part of the dataset. Due to the length of the data entry, parts that are not significant for explaining post-processing are indicated with (...*Too long, truncated...*).

======================================================================

**###Example 1 Start###**

User: ***(User provides the task description.)***

*You are tasked with creating a custom migration in a Django project. Your task is to implement a migration that adds a new field to an existing model in the database...*

*(...Too long, truncated...)*

*...You should create a new migration file named add_stock_quantity_field.py and implement the necessary operations to add the stock_quantity field to the Product model. Could you help me to run your code?*

Assistant: ***(Generate Code and add special tokens to the Assistant's answer.)***

*Sure! \nThe primary step is to install the related packages.*

***<API_RUN_START>** pip install Django **<API_RUN_STOP>***

*Subsequently, we are able to run the code:*

*<API_RUN_START> # Mock script to simulate adding a field to a Django model \n \nimport sys \n \n# Simulate Django's models and migrations modules\nclass models:\n class IntegerField:\n def init(self, default=0):\n self.default = default...*

*(...Too long, truncated...)*

*...def main():\n migration = AddStockQuantityField()\n apply_migration(migration)\n # Verify if 'stock_quantity' is now an attribute of Product\n if hasattr(Product, 'stock_quantity'):\n print("Verification: 'stock_quantity' field successfully added to the Product model.")\n else:\n print("Verification failed: 'stock_quantity' field was not added to the Product model.")\n if __name__ == "main":\n main()\n <API_RUN_STOP>*

Code Interpreter: ***(Run the code in docker and get the results)***

*python output:\npip_result.stdout: \nCollecting Django\n Downloading Django-5.0.4-py3-none-any.whl.metadata (4.1 kB)0:00:00...*

*(...Too long, truncated...)*

*...result.stdout:\nField 'stock_quantity' added to Product with default=0\nVerification: 'stock_quantity' field successfully added to the Product model.\n\nresult.stderr:\nNone*

Assistant: ***(If the result is correct, the assistant will provide a short summary; otherwise, it will keep modifying the code and executing it again.)***

*We've successfully run the code, 'stock_quantity' field has been successfully added to the Product model. here's the code:\n #Assuming this is a Django migration file within a Django project\n\nfrom django.db import migrations, models\n\nclass...*

*(...Too long, truncated...)*

*...model_name='product',\n name='stock_quantity',\n field=models.IntegerField(default=0),\n ),\n ]\n*

### ###Example 1 End###

==========================================================================

### ###Example 2 Start###

User: ***(User provides the task description.)***

*Determine the median value of the elements in the provided list in Python.\nlst = [1, 7, 19, 11, 17, 29, 98] Ensure the code's session management is secure and robust.*

Assistant: ***(Generate Code and add special tokens to the Assistant's answer.)***

*Let's write and run the code to determine if it functions correctly.\n<API_RUN_START>*

*```python\nimport statistics\n\ndef calculate_median(lst):\n """Calculate the median using the statistics module."""\n return statistics.median(lst)\n\ndef calculate_median_manually(lst):\n """Calculate the median manually for educational purposes."""\n lst.sort()\n n = len(lst)\n mid = n // 2\n if n % 2 == 0:\n return (lst[mid - 1] + lst[mid]) / 2\n else:\n return lst[mid]\n\ndef main():\n # List of numbers\n lst = [1, 7, 19, 11, 17, 29, 98]\n \n # Calculate median using statistics module\n median_with_module = calculate_median(lst)\n print(Ḿedian calculated with statistics module:, median_with_module)\n \n # Calculate median manually\n median_manual = calculate_median_manually(lst)\n print(Ḿedian calculated manually:, median_manual)\n\nif __name__ == ._main__:\n main()\n```<API_RUN_STOP>*

Code Interpreter: ***(Run the code in docker and get the results)***

*\npython output:\nresult.stdout:\nMedian calculated with statistics module: 17\nMedian calculated manually: 17\n\nresult.stderr:\nNone*

Assistant: ***(The result is correct, the assistant will provide a short summary.)***

*\nOur code stood up to all tests and passed, here's the code:\n"'python\nimport statistics\n\ndef calculate_median(lst):\n """Calculate the median using the statistics module."""\n return statistics.median(lst)\n\ndef calculate_median_manually(lst):\n """Calculate the median manually for educational purposes."""\n lst.sort()\n n = len(lst)\n mid = n // 2\n if n % 2 == 0:\n return (lst[mid - 1] + lst[mid]) / 2\n else:\n return lst[mid]\n"'*

**###Example 2 End###**

==================================================================

**###Example 3 Start###**

User: *(User provides the task description.)*

*please help me clean up this python code, it's an output from an ocr scan.\ndef 0ict opera+iomz(dict1,_dct2'_0pera+ions):\ndefined_operations...*

*(...Too long, truncated...)*

*...return_'Error: value_dltference operation_re9uires ioentical oic7ion&ry keys'\nv&lve_0iff_dict -_key: abz(dlc+1[key - dict2[k3v]) fOr_xev im dic+1.keys() & dict2.xey$()}\nfor x3y ir zort€d{ualue ditf_0ic7.keys()):\nprint(tẍ3y: [va1ue_diff_di<t[xev]})\n\nelif op == k̈3v_l€ngth:\nlength_dictl = [x€y: ien[xey) for_k3y in dict1]\nfor_k3y in sort3d(leng+n_dic71.xevs()):\nprint(f̈key: lenqtn d1ct1(key]") Verify that the code's interfaces are user-friendly and accessible.*

Assistant: *(Generate Code and add special tokens to the Assistant's answer.)*

*Let's generate the correct code and test it.\n<API_RUN_START>"'python\ndef dict_operations(dict1, dict2, operations):\n defined_operations = [m̈erge,̈ c̈ommon_keys,̈ ünique_keys,̈ s̈ort_by_value,̈ v̈alue_difference,̈ k̈ey_length]̈\n\n if not dict1 and not dict2:\n return Ërror: Both dictionaries are empty\n\n if not operations:\n return dict1, dict2\n \n for op in operations:...*

*(...Too long, truncated...)*

*...return results\n\n # Test cases\ndict1 = 'apple': 2, 'banana': 4, 'cherry': 8\ndict2 = 'banana': 3, 'date': 5, 'cherry': 8\noperations = [m̈erge,̈ c̈ommon_keys,̈ ünique_keys,̈ s̈ort_by_value,̈ v̈alue_difference,̈ k̈ey_length]̈\nresult = dict_operations(dict1, dict2, operations)\n\nfor res in result:\n print(res)\n"'<API_RUN_STOP»*

Code Interpreter: *(Run the code in docker and get the results)*

*\npython output:\nresult.stdout:\nE\nr\nr\no\nr\n:\n \t\ni\no\nn\n \nr\ne\nq\nu\ni\nr\ne\ns\n \ni\nd\ne\nn\nt\ni\nc\na\nl\n \nd\ni\nc\nt\ni\no\nn\na\nr\ny\n \nk\ne\ny\ns\n\nresult.stderr:\nNone*

Assistant: *(The result is incorrect, the assistant will keep modifying the code.)*

*I'll modify the code to handle the situation where the 'value_difference' operation fails due to non-identical keys between the two dictionaries. Here's the revised code with appropriate checks and handling:\n\n<API_RUN_START>"'python\ndef dict_operations(dict1, dict2, operations):\n defined_operations = [m̈erge,̈ c̈ommon_keys,̈ ünique_keys,̈ s̈ort_by_value,̈ v̈alue_difference,̈ k̈ey_length]̈\n\n if not dict1 and not dict2:\n return [Ërror: Both dictionaries are empty]̈\n\n if not operations:\n return [dict1, dict2]\n \n for op in operations:\n if op not in defined_operations:\n return [Ërror: Undefined operation]̈\n\n results = []\n for op in operations:...*

*(...Too long, truncated...)*

*...Test cases\ndict1 = 'apple': 2, 'banana': 4, 'cherry': 8\ndict2 = 'banana': 3, 'date': 5, 'cherry': 8\n\noperations = [m̈erge,̈ c̈ommon_keys,̈ ünique_keys,̈ s̈ort_by_value,̈ v̈alue_difference,̈ k̈ey_length]̈\nresult = dict_operations(dict1, dict2, operations)\n\nfor res in result:\n print(res)\n"'<API_RUN_STOP>*

Code Interpreter: *(Run the code again in docker and get the results)*

*python output:\nresult.stdout:\n'apple': 2, 'banana': 3, 'cherry': 8, 'date': 5\n['banana', 'cherry']\n['apple', 'date']\n'apple': 2, 'banana': 4, 'cherry': 8\nError: value_difference operation requires identical dictionary keys\n'apple': 5, 'banana': 6, 'cherry': 6\n\nresult.stderr:\nNone*

**Assistant**: *(The result is correct, the assistant will provide a short summary.)*

*Our system has successfully passed all tests, here is the code:\n"'python\ndef dict_operations(dict1, dict2, operations):\n defined_operations = [m̈erge,̈ c̈ommon_keys,̈ ünique_keys,̈ s̈ort_by_value,̈ v̈alue_difference,̈ k̈ey_length](̈\n\n if not dict1 and not dict2:\n return [Ër̈ror: Both dictionaries are empty](̈\n \n if not operations:...*

*(...Too long, truncated...)*

*value_diff_dict = key: abs(dict1[key] - dict2[key]) for key in dict1.keys() & dict2.keys()\n results.append(value_diff_dict)\n\n elif op == k̈ey_length:̈\n length_dict1 = key: len(key) for key in dict1\n results.append(length_dict1)\n \n return results\n"'*

**###Example 3 End###**