# OpenReview forum: "AutoCoder: Enhancing Code Large Language Model with AIEV-INSTRUCT"
_ICLR.cc/2025/Conference — Submitted to ICLR 2025_

### Official Review · Reviewer_etyE · 2024-10-17

**Soundness:** 2
**Presentation:** 2
**Contribution:** 2
**Rating:** 3
**Confidence:** 4

**Summary:**

This paper introduces a framework to synthesize datasets of code generation with two steps: 1) distilled from the teacher model (GPT-4) by implementing two-agent framework with the extra execution platform; 2) student model self-learning when its capabilities are stronger or comparable to the teacher model. In both stages, no human annotation is involved, which may cause some issues. Then the authors fine-tune the DeepSeek models with the created dataset and achieve comparable performance to GPT-4 and Claude-3.5 on coding tasks such as HumanEval and MBPP. One critical point is that the authors consider the limitation of current Code Interpreter in OpenAI that does not have the strategy to install required packages during task execution. Hence, the authors modify the dataset to better align this capability by implementing the bash commands.

While this paper does have some great points like considering Code Interpreter, the writing is poor with many syntax errors (seemingly written in a rush) and some critical questions of why the framework can work is not addressed. I recommend a rejection in the current stage. Though with negative commands, I still appreciate some of the contributions of authors, and strongly recommend authors to better polish the paper for next submissions.

**Strengths:**

The creation of the dataset with purely LLM based methods is appreciated. The authors try to avoid the efforts of human annotation, which is a reasonable and important.

The consideration of Code Interpreter is great! This is an important function in OpenAI GPT models. However, current code LLMs rarely consider this feature. The authors also try to fix the gap that current GPT Code Interpreter cannot install packages.

The finetuning on LLMs with 33B parameters shows much efforts and resources the authors spend. The model turns out to achieve comparable performance.

**Weaknesses:**

The paper seems being written in a rush and has many syntax and expression errors.
Line 36 and 37, economically and time-consuming; line 64, sperate -> separate; line 65 and 66, proprietary -> proper; line 93 boast -> boost; Figure 3, If the teacher model have better performance -> If the teacher model has better performance, the student model have -> the student model has; Line 67, Self-learning -> Self-Learning; Line 215, programmer , -> programmer; Line 265 Figure 5, Nature language -> Natural language; Line 351, We -> we; Line 378, large language models -> Large Language Models; Line 533, teaching stage -> Teaching stage, self-learning stage -> Self-Learning stage;

1)	How to handle the case that the programmer cannot synthesize the correct code after multiple rounds of iteration? Purely discarding the unsuccessful responses will introduce much bias into dataset.
2)	Following the above question, how to make sure the final synthesized coding questions and required codes possess enough diversity? Some coding questions may be too hard for GPT-4o to handle. Without human annotation, the synthesized dataset may have bias into simple and specialized questions.
3)	The tested coding tasks are not hard enough. How about SWE-bench? Tasks like HumanEval have errors in some questions. Accuray difference of 2-5% may come from the dataset errors.
4)	What is the intuition that the student model can perform better than the teacher model even just trained on the synthesized dataset without human annotation?
5)	In the Self-Learning stage, the student model iterates by itself. Then how to make sure it can be better not even worse? For instance, some errors will be accumulated during dataset creation without human annotation. Some codes will be judged as correct but wrong. Some hard coding problems will never be generated since the model itself cannot iterate into this ability.
6)	The results in Table 1 show nearly no improvements to the teacher model GPT-4. The proposed framework still seems as another kind of knowledge distillation from the larger model.
7)	The theoretical analysis to prove better performance in Appendix Section B is not solid. For example, the assumption that iterative validation increases correctness is not such ideal. The error can appear in all the iteration steps, not decreasing over rounds.
8)	Figure 3 and Figure 5 are not clear and easy for readers to understand.

**Questions:**

The questions have been listed in the Weakness section above.

---

### Official Review · Reviewer_AxAo · 2024-10-25

**Soundness:** 3
**Presentation:** 2
**Contribution:** 3
**Rating:** 5
**Confidence:** 3

**Summary:**

This paper proposes Autocoder, trained by fine-tuning existing LLMs on a dataset called AIEV-Instruct. AIEV stands for Agent-Interaction and Execution-Verified, which involves two LLM-based agents (namely, Programmer and Questioner) generating multi-turn dialogue with the help of a code interpreter. The curation of AIEV consists of one stage of collecting data from closed-source models and another stage using self-trained models, resulting in 169K data samples. The final Autocoder models show promising performance on code generation benchmarks such as HumanEval and MBPP.

**Strengths:**

Overall, the framework is rather straightforward and technically makes sense. models trained on AIEV also show decent improvements and outperform many existing open-source models.

**Weaknesses:**

1. Minor typos in the paper: the authors should use \citep{} for most of the citations.
2. I suggest the authors make more organized presentations in Figure 3 and Figure 4, and clearly state the process of how AIEV data is curated in the caption. This will make the method easier to follow and correlate with the text description in the main body.
3. Baselines comparison:
   - For Table 1, I wonder if the authors could provide results on tuning OSS-Instruct, i.e., Magicoder datasets?
   - For Qwen baselines, e.g., Qwen-2.5 in Table 1, I wonder if the authors could include their code-specific model, i.e., Qwen-2.5-Coder-Instruct, rather than the general model Qwen-2.5-Instruct?
   - The experiment section can be clearer if the authors provide some baseline setup at the beginning of Section 5 or state them more explicitly.
4. Clarification for Sec 5.5 and Figure 6:
   - The authors should explain what ST and MT datasets they are using. Are these datasets publicly available? Or is Figure 6 indeed an ablation study of the AIEV methodology (source data, MT dataset, and MT with execution)?

**Questions:**

1. In the demo video, the failure of GPT-4o to install a Python package is very likely due to some safety or system security considerations, so the installation of any package is abandoned. Is there any head-to-head comparison demonstrating that Autocoder really excels in such a scenario?
2. I believe many agent frameworks integrating LLMs can support this function, such as [1][2][3][4], etc. It depends on how much users trust the LLMs to perform these risky operations rather than how powerful the code models are. Based on my understanding, the success of package installation appears to be more attributable to engineering efforts and the specific features present in the dataset, rather than a result of the model exhibiting a significantly superior emergent capability. While I acknowledge the authors' contributions, I suggest that this aspect of the work could be framed more objectively, highlighting the technical implementation as a key contribution rather than implying a sudden leap in model ability.

[1] AutoGPT, https://github.com/Significant-Gravitas/AutoGPT

[2] Open Interpreter, https://github.com/OpenInterpreter/open-interpreter

[3] OpenAgents: An Open Platform for Language Agents in the Wild, Xie T, Zhou F, Cheng Z, et al., 2024

[4] Agents: An Open-source Framework for Autonomous Language Agents, Zhou W, Jiang Y E, Li L, et al., 2024

---

### Official Review · Reviewer_WxvX · 2024-11-04

**Soundness:** 2
**Presentation:** 2
**Contribution:** 3
**Rating:** 3
**Confidence:** 4

**Summary:**

- This paper proposes a new data generation method for code.
- The goal of the authors is to address two problems in previous data generation methods.
- The first problem is that previous data generation methods for code sometimes generate incorrect solutions, and not enough of an effort is made to correct them.
- The second problem is that since we are distilling from a teacher, the accuracy of the student can never surpass the teacher.
- To address the first problem, the authors propose a data generation pipeline for code called AIEV-Instruct.
- AIEV instruct is an extension of OSS-Instruct that also asks the LLM to generate unit tests and has multiple rounds of dialogue using those unit tests. This lets the authors prevent bad solutions from slipping through.
- To address the second problem, the authors do extended self-training once the accuracy of the student surpasses that of the teacher.
- The authors also extend the code interpreter function with the ability to execute bash commands so the LLM can install packages.

**Strengths:**

- Addresses a real problem in the data generation pipeline for code generation.
- I like the idea of generating unit tests and using them to filter for code correctness.
- The proposed self-training method is an interesting hypothesis to test.
- Empirical results are strong and experimental evaluation is broad. Good job by the authors here.

**Weaknesses:**

- **Major Flaw**: I don't see any _direct_, _fair_ comparisons presented to previous data in a clear, transparent way.
    - L220: You seed data generation with Magicoder-Evol-Instruct and Magicoder-OSSInstruct.
    - **To show that your data generation method is better than simply training on Magicoder-Evol-Instruct and Magicoder-OSSInstruct**, you need to train a model on Magicoder-Evol-Instruct and Magicoder-OSSInstruct and compare it to the same model trained with the same settings on AutoCoder.
- L100: Claiming that "As far as we know, as of September 2024, AutoCoder is the only model that supports automatically installing external packages in the Code Interpreter" seems incorrect.
    - Any code interpreter that supports Jupyter notebooks (e.g. [OpenCodeInterpreter](https://github.com/OpenCodeInterpreter/OpenCodeInterpreter/blob/main/demo/code_interpreter/JupyterClient.py) or [CodeAct](https://github.com/xingyaoww/code-act?tab=readme-ov-file#start-your-code-execution-engine)) trivially supports arbitrary shell commands by emitting `! <any shell commands here>`.
- I do not see any proof the self-training (a main claimed contribution) works.
    - To demonstrate that the self-training works, you would need to show some plot or table that shows accuracy before the self-training and accuracy after the self-training. By accuracy, I mean accuracy on some of the benchmarks you are testing.
    - For example, take a look at Figure 6 in [this paper](https://openaccess.thecvf.com/content/CVPR2024/html/Khan_Self-Training_Large_Language_Models_for_Improved_Visual_Program_Synthesis_With_CVPR_2024_paper.html).

# Summary
Each of the claimed contributions has problems, some of them severe. You have not showed that your data is better in a fair comparison, and there is no evidence (as far as I can tell) that the self-training works.

My review may seem very harsh, but I like the paper and the claimed contributions. I do not see evidence to support the claimed contributions. This surprised me, given that the empirical evaluation is otherwise extensive. Right now, the main contribution is that you have a very good OSS coding model — which is great — but you have not provided any evidence that the OSS model is _great because of your pipeline_ and not because you trained a strong model on Magicoder-Evol-Instruct + Magicoder-OSSInstruct.

**Questions:**

1. I do not see a direct comparison between the Magicoder-Evol-Instruct + Magicoder-OSSInstruct data and the AIEV-Instruct data. If you do a direct comparison (same model, same training settings) can you state where you do it? If you cannot do it (even with the 7b model), can you explain why?

2. Can you explain why your CodeInterpreter is an improvement over the Jupyter notebook approach taken by works like OpenCodeInterpreter and CodeAct, which to me seem to have the same (and more) capabilities?

3. Can you provide any evidence the self-training works? If not, why? If you have already provided it and I have missed it, where is it?

**I want to be fair to the authors — if you think I am fundamentally misunderstand your presentation of results and somewhere you have shown that simply training DS-Coder on Magicoder-Evol-Instruct + Magicoder-OSSInstruct is clearly not as good as training DS-Coder on AIEV-Instruct, can you point that out to me?** For example, maybe it is in one of the tables — is one of the comparison models in fact DS-Coder trained on these datasets?

---

### Official Review · Reviewer_2ckE · 2024-11-04

**Soundness:** 2
**Presentation:** 1
**Contribution:** 2
**Rating:** 3
**Confidence:** 4

**Summary:**

This paper proposes AIEV-INSTRUCT pipeline and the corresponding model AutoCoder, and show improvement by fine-tuning Deepseek-Coder on it.

**Strengths:**

They provide some details of the experiments, such as the GPT version and the machine used, which are helpful for reproducibility.

**Weaknesses:**

1. The writing and presentation are unclear and inefficient. Although the paper includes some details, most fail to offer a real understanding of the core algorithm, and some important information is still missing. Specifically, how is the data generated? Is it directly from the uncontaminated data you mentioned, with questions written based on the code? What prompts are used? Regarding the figures, most appear more as advertisements aiming to grab attention than thoughtfully designed visual aids. The fonts are oversized, and in Figure 3, lines 162-170 and 172-181 are exactly repeated.

2. I'm not sure what is the novelty based on the unclear presentation. Is this more efficient then previous methods? The writing seems to only focus on the engineering implementation.

3. Missing baselines. To demonstrate that Auto-Instruct outperforms previous methods like Evol-Instruct, experiments should include fine-tuning the same model, such as DeepseekCoder, on datasets generated by various methods with comparable size and generation cost, considering rewriting by GPT-4. However, no such results are provided.

**Questions:**

See weakness section. What are the real difference between Auto-Instruct and previous methods? Are they based on a known dataset and rewrite some parts and concat the results obtained by a Python environment? What are the prompts? Can you make all these clear without inefficient figures and words?

---

### Meta-Review · Area_Chair_Tpmu · 2024-12-21

**Metareview:**

The paper introduces AutoCoder, an open-source Large Language Model (LLM) designed for code generation, which surpasses GPT-4 Turbo and GPT-4o in pass@1 on the Human Eval benchmark (90.9% vs. 90.2%). AutoCoder leverages a novel data generation pipeline termed AIEV-Instruct (Agent-Interaction Execution-Verified), which combines agent interactions and external code execution verification to create a multi-turn dialogue dataset. This approach aims to reduce reliance on proprietary models and enhance data accuracy. The model also features an advanced code interpreter capable of installing external packages, a feature not available in GPT-4 Turbo and GPT-4o. The authors fine-tune DeepSeek-Coder using the AIEV-Instruct dataset and demonstrate competitive performance on several code generation benchmarks.

#### Contribution
1. **AIEV-Instruct Pipeline**: A novel method for generating high-quality code datasets using agent interactions and execution verification, reducing reliance on proprietary models.
2. **AutoCoder Model**: An open-source LLM that outperforms GPT-4 Turbo and GPT-4o on the Human Eval benchmark, offering a versatile code interpreter with package installation capabilities.
3. **Empirical Validation**: Comprehensive benchmarking on code generation tasks (e.g., HumanEval, MBPP) showcasing AutoCoder's competitive performance.

#### Weaknesses
1. **Unclear Presentation**: The paper suffers from poor writing and presentation, with repetitive and inefficient descriptions that obscure the core contributions. Figures are criticized as being more promotional than informative.
2. **Lack of Novelty Clarity**: The novelty of the AIEV-Instruct pipeline relative to existing methods (e.g., Evol-Instruct, OSS-Instruct) is not clearly articulated, raising questions about its efficiency and improvements over prior work.
3. **Missing Baselines**: The absence of direct, fair comparisons with existing datasets (e.g., Magicoder-Evol-Instruct, Magicoder-OSSInstruct) undermines claims of superiority. The paper does not fine-tune the same model on these datasets for comparison.
4. **Unsubstantiated Claims**: Assertions about AutoCoder's unique ability to install external packages are questioned, as similar capabilities exist in other frameworks. The effectiveness of self-training, a core contribution, lacks empirical evidence.
5. **Dataset Bias and Diversity**: Concerns are raised about potential biases in the synthesized dataset due to the lack of human annotation, and the diversity of generated coding tasks is questioned.
6. **Benchmark Limitations**: The tasks evaluated (e.g., HumanEval, MBPP) are considered insufficiently challenging, and the paper does not address more complex benchmarks like SWE-bench.
7. **Theoretical Analysis**: The theoretical justification for improved performance in the appendix is deemed weak, with unrealistic assumptions about iterative validation reducing errors.

**Additional Comments On Reviewer Discussion:**

The rebuttal period for "AutoCoder: Enhancing Code Large Language Model with AIEV-INSTRUCT" involved no recorded responses or interactions between the authors and reviewers. Consequently, the paper's content remains unchanged from the initial submission, and the reviewers' concerns have not been addressed.

**Points Raised by Reviewers:**
1. **Unclear Presentation (2ckE, etyE):**
   - Reviewers criticized the paper for poor writing, syntax errors, and inefficient figures. Suggestions included improving clarity, fixing typos, and redesigning figures to better convey the methodology.

2. **Lack of Novelty Clarity (2ckE, WxvX):**
   - Reviewers questioned the novelty of AIEV-Instruct, asking for a clearer comparison with existing methods. They sought details on data generation, prompt usage, and the pipeline's efficiency.

3. **Missing Baselines (2ckE, WxvX, AxAo):**
   - Multiple reviewers emphasized the need for direct comparisons with existing datasets (e.g., Magicoder-Evol-Instruct) by fine-tuning the same model (e.g., DeepSeek-Coder) under identical conditions.

4. **Unsubstantiated Claims (WxvX, AxAo):**
   - The claim of AutoCoder being the only model to support external package installation was disputed, with reviewers noting similar capabilities in other frameworks. The self-training approach lacked empirical validation.

5. **Dataset Bias and Diversity (etyE):**
   - Concerns were raised about the potential biases introduced by omitting unsuccessful code synthesis attempts and the lack of diversity in generated tasks without human annotation.

6. **Benchmark Limitations (etyE):**
   - The chosen benchmarks were deemed insufficiently challenging, with suggestions to include more rigorous tests like SWE-bench to validate AutoCoder's performance.

7. **Theoretical Analysis (etyE):**
   - The theoretical justification for improved performance was criticized for making unrealistic assumptions about error reduction through iterative validation.

**Final Decision Weighting:**
The reviewers' feedback highlights significant shortcomings in the paper's presentation, methodology, and empirical validation. The lack of clarity in describing the AIEV-Instruct pipeline, the absence of proper baselines, and unsubstantiated claims undermine the paper's contributions. Additionally, the lack of author responses during the rebuttal period suggests a missed opportunity to address these concerns.

Given the unresolved issues and the reviewers' consensus leaning towards rejection, the final decision is to **reject** the paper. The authors are encouraged to address the highlighted weaknesses, particularly by improving the presentation, providing fair comparisons with existing methods, and validating their claims with rigorous empirical evidence, before resubmission.

---

### Decision · Program_Chairs · 2025-01-22

Reject